# Unobtrusive Skin Temperature Estimation on a Smart Bed

**DOI:** 10.3390/s24154882

**Published:** 2024-07-27

**Authors:** Gary Garcia-Molina, Trevor Winger, Nikhil Makaram, Megha Rajam Rao, Pavlo Chernega, Yehor Shcherbakov, Leah McGhee, Vidhya Chellamuthu, Erwin Veneros, Raj Mills, Mark Aloia, Kathryn J. Reid

**Affiliations:** 1Sleep Number Labs, San Jose, CA 95113, USA; trevor.winger@sleepnumber.com (T.W.); nikhil.makaram@sleepnumber.com (N.M.); megharajam.rao@sleepnumber.com (M.R.R.); vidhya.chellamuthu@sleepnumber.com (V.C.); erwin.veneros@sleepnumber.com (E.V.); 2Department of Psychiatry, University of Wisconsin, Madison, WI 53719, USA; 3Department of Computer Science and Engineering, University of Minnesota, Minneapolis, MN 55455, USA; 4GlobalLogic, 03038 Kyiv, Ukraine; pavlo.chernega@sleepnumber.com (P.C.); yehor.shcherbakov@sleepnumber.com (Y.S.); 5Sleep Number Corporation, Minneapolis, MN 55404, USA; leah.mcghee@sleepnumber.com (L.M.); rajasi.mills@sleepnumber.com (R.M.); mark.aloia@sleepnumber.com (M.A.); 6Department of Medicine, National Jewish Health, Denver, CO 80206, USA; 7Center for Circadian and Sleep Medicine, Department of Neurology, Feinberg School of Medicine, Northwestern University, Chicago, IL 60611, USA

**Keywords:** unobtrusive, sleep, skin temperature, regression model, temperature sensor strip, smart bed

## Abstract

The transition from wakefulness to sleep occurs when the core body temperature decreases. The latter is facilitated by an increase in the cutaneous blood flow, which dissipates internal heat into the micro-environment surrounding the sleeper’s body. The rise in cutaneous blood flow near sleep onset causes the distal (hands and feet) and proximal (abdomen) temperatures to increase by about 1 °C and 0.5 °C, respectively. Characterizing the dynamics of skin temperature changes throughout sleep phases and understanding its relationship with sleep quality requires a means to unobtrusively and longitudinally estimate the skin temperature. Leveraging the data from a temperature sensor strip (TSS) with five individual temperature sensors embedded near the surface of a smart bed’s mattress, we have developed an algorithm to estimate the distal skin temperature with a minute-long temporal resolution. The data from 18 participants who recorded TSS and ground-truth temperature data from sleep during 14 nights at home and 2 nights in a lab were used to develop an algorithm that uses a two-stage regression model (gradient boosted tree followed by a random forest) to estimate the distal skin temperature. A five-fold cross-validation procedure was applied to train and validate the model such that the data from a participant could only be either in the training or validation set but not in both. The algorithm verification was performed with the in-lab data. The algorithm presented in this research can estimate the distal skin temperature at a minute-level resolution, with accuracy characterized by the mean limits of agreement [−0.79 to +0.79 °C] and mean coefficient of determination $R^{2}=0.87$. This method may enable the unobtrusive, longitudinal and ecologically valid collection of distal skin temperature values during sleep. Therelatively small sample size motivates the need for further validation efforts.

## 1. Introduction

Sleepiness and vigilance are closely associated with core body (CBT) and skin temperatures. A high daytime CBT aligns with a low skin temperature for optimal vigilance, while a low night-time CBT aligns with a high skin temperature for optimal sleep. Sleep initiation is bidirectionally associated with the circadian rhythms of the CBT and skin temperature, with habitual sleep onset coinciding with CBT decline and increased skin temperature [1].

The CBT decline during sleep is facilitated by peripheral vasodilation, which increases the cutaneous blood flow such that internal heat from the body’s core is dissipated into the micro-environment surrounding the sleeper’s body. The rise in blood flow through peripheral cutaneous blood vessels causes the skin temperature to rise [2]. This is particularly evident during the transition from wakefulness to sleep, which is accompanied by an increase in both the distal (hands and feet) and proximal (abdomen and chest) skin temperature by approximately 1 °C and 0.5 °C, respectively [3].

Under the controlled conditions of a constant routine protocol, Kräuchi et al. [3] identified the gradient between the distal and proximal skin temperature to be the best predictor for sleep onset latency. A causative role of warming the feet in increasing the distal skin temperature and shortening the sleep onset latency was verified in [4,5]. The warming of the proximal temperature by less than 1 °C can reduce the sleep onset latency by at least 3 min, as reported in [6].

The close association between sleep, body temperature regulation, and patterns of skin temperature change motivates the need for practical and ecologically valid means to unobtrusively measure the skin temperature during sleep and enable the longitudinal characterization of the dynamics of skin temperature during sleep.

The research presented in this article proposes an algorithm to estimate the wrist distal skin temperature during sleep with a minute-level temporal resolution using the data from a temperature sensor strip positioned slightly under the surface of a smart bed’s mattress.

## 2. Materials and Methods

### 2.1. Smart Bed Platform

Sleep Number’s primary sensing technology uses a pressure signal sampled at 500 Hz from a sensor positioned inside an inflatable air bladder within a smart bed. Ballistocardiography (BCG) signals, which reflect movement, cardiac activity, and respiratory activity, are obtained from the pressure signal through signal processing methods described in [7]. The smart bed’s embedded software relies on machine learning methods to determine, in real time, bed presence, bed entry, bed exit, body movements, position changes, time to fall asleep, breathing rate, and heart rate [7].

In addition to the pressure signal, recently introduced smart bed models (C360 [8]) include an array of five temperature sensors arranged along a temperature sensor strip (TSS). These sensors are impalpable as they are located 0.25 inches below the sleeping surface.

The C360 smart bed type used in this study has two TSSs on each side of the bed, positioned as indicated in Figure 1. For a queen-sized bed (width: 152.4 cm height: 203.2 cm), the individual temperature sensors are separated from each other by 14 cm. For a king-sized bed (width 203.2 cm and height: 203.2 cm), the spacing between the individual temperature sensors is 16.8 cm.

The individual temperature sensors are referred to as T1 to T5 starting from the edge to the center of the bed. The embedded software in the smart bed samples the temperature data of each individual sensor at an approximate rate of 1 sample/s. The resolution of each temperature sensor is 0.1 °C.

### 2.2. Study Design and Participants

This study, approved by the Institutional Review Board (IRB) of Northwestern University (protocol number STU00217800), was conducted between August 2023 and March 2024. Eighteen volunteers were recruited (9M/9F), with a mean age of 44.5 (standard deviation SD: 6.7) years (see the table in Section 3.1), from Sleep Number customers having a C360 smart bed and residing within easy reach of the Sleep Health Center of Northwestern University in the western Chicago area, IL, USA.

The study volunteers signed an IRB-approved consent form and participated in this study for 16 nights in total. Each participant collected data from 14 sleep sessions at home in their usual smart bed and spent two nights in the sleep lab on a queen-sized C360 smart bed. Two queen-sized C360 smart beds, to which participants were randomly assigned, were installed in the research facilities at the Sleep Health Center at Northwestern University.

The in-home portion of the study included the collection of data from the TSS; the proximal skin temperature, using an iButton [9] (type DS1922L; Maxim, Dallas Semiconductor Corp., Dallas, TX, USA) positioned just below the left collar bone on the participant’s chest; the foot distal skin temperature, using an iButton positioned near the arch on the right foot; and the non-dominant wrist distal skin temperature, using the Empatica E4 (Boston, MA, USA) wristband [10]. The iButton device offers two resolutions, 0.5 °C or 0.0625 °C [11]; we used the latter in this study. The Empatica E4’s skin temperature resolution is 0.02 °C [12]

The in-lab portion of the study included polysomnography (PSG) signals [13], the proximal skin temperature, and the foot distal skin temperature, measured using iButtons positioned similarly to the in-home condition. The Empatica wristband was not used in the lab to minimize the burden on participants who slept with a PSG setup that included an electroencephalogram, electrocardiogram, electromyogram, and breathing sensors. The TSS data from the smart beds installed in the sleep lab were also collected.

The sampling periodicity for the iButtons was set to 30 s, the sampling rate for the skin temperature acquired by the Empatica wristband was 4 samples/s, and the TSS sampling rate was approximately 1 sample/s/sensor.

In this research, we focus on estimating the wrist distal skin temperature (i.e., the temperature measured by the Empatica wristband) using the TSS data. Henceforth, we refer to the wrist distal skin temperature as DST.

The analysis of the complete dataset collected in this study including PSG signals, sleep architecture, and behavioral outcomes is ongoing and will be the topic of a future publication.

### 2.3. Distal Skin Temperature Estimation Algorithm

Figure 2 shows the block diagram of the algorithm to estimate the DST at time *n* in minutes. An early version of this algorithm was presented in [14].

The data from the temperature sensor array, acquired at a sampling frequency of approximately a sample per second for each sensor, were aggregated at the minute level by calculating the mean value such that the sampling rate for each individual temperature sensor was exactly 1 sample/min. The temperature data from the Empatica wristband and the iButtons were also sub-sampled at 1 sample/min to match the TSS’s sampling rate. Given the low variability of the skin temperature values [15], such an aggregation can be performed without the risk of aliasing.

The five TSS values for the n-th minute $T_{1}\left(n\right),...,T_{5}\left(n\right)$ were sorted in descending order $T_{i1}\left(n\right)\geq T_{i2}\left(n\right)\geq T_{i3}\left(n\right)\geq T_{i4}\left(n\right)\geq T_{i5}\left(n\right)$; i1,...,i5∈{1,...,5} and the top three largest values $V_{1}\left(n\right)=T_{i1}\left(n\right),V_{2}\left(n\right)=T_{i2}\left(n\right),V_{3}\left(n\right)=T_{i3}\left(n\right)$ were selected for further processing. This selection aimed at focusing the analysis on the temperature values that most likely touched the body of the study participant.

A first DST estimation X(n) was obtained using a gradient boosted tree implemented using the XGboost library in Python [16] $X\left(n\right)=\mathrm{XGboost}(V_{1}\left(n\right),V_{2}\left(n\right),V_{3}\left(n\right))$. Because of the strong correlation between consecutive DST values [15], five consecutive DST estimates were grouped into a five-element vector X(n),...,X(n−4), which was further processed by a random forest [17] to produce the final DST estimation at time *n*, Y(n)=RandomForest(X(n),...,X(n−4)). The two regression models, namely, the gradient boosted tree and random forest, were sequentially trained as described in Section 2.4.

### 2.4. Training and Validation of the Gradient Boosted Tree and the Random Forest Models

The data from all the sleep sessions from the in-home portion of the study were used to train and validate the regression models. Data selection criteria were applied to ensure that only high-quality data were used to train the models. The data from a sleep session were included if at least 80% of the TSS and Empatica data for that session had good enough quality characterized by the absence of missing data or invalid values that included null, negative, and large temperature oscillations within a minute (which indicate electronic issues). In addition, using the smart bed data from the pressure signal, the temperature data corresponding to segments of bed absence were discarded.

All the sleep session data from a given participant were considered as an indivisible block such that no data from the same participant could be in both training and validation sets. This procedure increases the likelihood of the validation accuracy to reflect the generalization accuracy (i.e., the accuracy on unseen data).

The skin temperature data measured at the non-dominant wrist by the Empatica E4 device was set as the ground-truth (or reference) DST value. These data were first smoothed using the robust locally weighted regression algorithm (LOWESS [18]) and then sub-sampled from 4 samples/second to 1 sample/min to match the TSS sampling rate.

A five-fold cross-validation (CV) procedure [19] was followed to train the regression models. The set of participants was partitioned into five subsets that were mutually exclusive and collectively exhaustive. For each CV fold, the data from a given subset were selected for accuracy evaluation (see accuracy evaluation method in Section 2.5) and the data from the remaining four subsets were used to train the regression models in sequential order: the gradient boosted tree first, and then the random forest (see Figure 3a).

The optimal set of hyperparameters for both regression models was determined by selecting a portion (20%) of the data to perform a Bayesian-type of optimization [20] (implemented in Python using the Hyperopt library) in the hyperparameter space defined as described in Table 1. The optimal hyperparameter values were used for the five-fold CV.

### 2.5. Accuracy Evaluation

For each CV fold, the accuracy was evaluated at the sleep session level first (see Figure 3b) and then the mean values across all sleep sessions in the validation set were calculated to obtain the accuracy corresponding to the CV fold.

At the sleep session level, the agreement between the ground-truth values obtained from the reference DST and the estimated DST values (smoothed using LOWESS) was evaluated using Bland–Altman plots [21], from which the bias, the upper limit of agreement (LoA), and the lower LoA were calculated. Given the ground-truth DST values for a sleep session S(5),...,S(N) and the corresponding DST estimated values Y(5),...,Y(N), the bias, lower LoA, and upper LoA were calculated as described in Equation (Equation 1). In this equation, the initial value for *n* is 5 since the first estimation Y(n) can only occur after five values: X(n−4),...,X(n) have been accumulated (see Figure 2). $SD_{d}$ is the standard error and the limits of agreement LoA correspond to the the 95% confidence interval for the bias.
(1)$$\begin{array}{cc} \mathrm{Bias} & =\frac{1}{N}\sum_{n=5}^{N}\left(S(n)-Y(n)\right),\\ SD_{d} & =\sqrt{\frac{1}{N-5}\sum_{n=5}^{N}{\left(S\left(n\right)-Y\left(n\right)-\mathrm{Bias}\right)}^{2}},\\ \mathrm{Lower}\;\mathrm{LoA} & =\mathrm{Bias}-1.96\times SD_{d},\\ \mathrm{Upper}\;\mathrm{LoA} & =\mathrm{Bias}+1.96\times SD_{d}. \end{array}$$

In addition to the bias and LoA, the coefficient of determination $R^{2}$ was also calculated to quantify the agreement between the reference and estimated DST. The coefficient of determination is informative of the regression type of analyses [22] and can be interpreted as the proportion of the variation in the reference DST that is explained by the estimated DST.

### 2.6. DST Estimation Algorithm Verification Using the in-Lab Data

The DST for all in-lab sessions was estimated using the algorithm presented in Section 2.3. For each sleep session, the estimated DST values starting at lights-off (which is the event that indicates to the study participants that they may start attempting to fall asleep) and ending an hour after lights-off, were normalized by subtracting the initial value to obtain a DST change curve.

The hour-long period for this analysis was intended to capture the falling asleep process which should be associated with an increase in DST, as elaborated in Section 1. The grand average estimated DST change across all in-lab sessions was finally calculated by averaging the individual sleep session DST change curves.

An identical process to the one described in the previous paragraph was followed for the measured foot distal temperature data to obtain a grand average foot distal temperature change (with respect to lights-off) across all in-lab sessions. While the latter may not be identical to the grand average estimated DST change because of the difference in location (wrist versus foot), it is reasonable to expect a significant and strong correlation between these curves during the falling asleep process. We have therefore calculated the Pearson correlation and corresponding statistical significance.

## 3. Results

### 3.1. Dataset for the Analysis

Two of of the eighteen participants (S17 and S18 in Table 2) in the study completed only the in-lab portion of the study. The remaining 16 participants completed both the in-lab and in-home portions. Table 2 shows the demographic information of the participants in this study.

The data from the in-home portion were used for training and validation. From the 224 (=16×14) in-home sleep sessions, the data from 49 sleep sessions were not considered for further analysis due to insufficient quality of more than 20% of the TSS data from a sleep session (see Section 2 for the quality criteria). From the remaining 175 sleep sessions, 85 sessions were discarded due to insufficient quality of more than 20 % of the Empatica temperature data from a sleep session.

### 3.2. Training and Validation

Through the implementation of the hyperparameter optimization procedure described in Section 2.4, the optimal hyperparameter values listed in Table 3 were identified. These were used in the five-fold CV procedure.

The learning curves in Figure 4, show the mean coefficient of determination across the five CV folds for the training and validation sets versus the percentage of training data used. As the amount of training data increases, the gap between training and validation diminishes, which suggests the appropriate generalization ability of the DST estimation model.

The mean accuracy values across the sleep sessions in the validation set for all the CV folds are reported in Table 4. The mean coefficient of determination is $R^{2}=0.87$ (SD: 0.03), indicating that almost 90% of the variability in the DST is explained by the estimated DST. The mean bias is −0.002 °C (SD: 0.008 °C), mean lower LoA =−0.79 °C (SD: 0.11 °C), and upper LoA =0.79 °C (SD: 0.11 °C). Thus, the absolute value of 95% of the DST approximation errors are within a 1 °C interval.

The Bland–Altman plots corresponding to each CV fold reported in Table 4 are shown in Figure 5. The bias is quasi equal to zero and the limits of agreement characterizing the 95% confidence interval around the bias are within the −1 to +1 °C interval.

Nine examples of ground-truth and estimated DST curves from sleep sessions randomly chosen among the ones in the validation set are represented in Figure 6. These illustrate the ability of the algorithm to capture the ground-truth (Empatica) DST variability with a low bias. These data were collected during the in-home sessions and the reference time corresponds to bed entry.

### 3.3. Verification Using the in-Lab Data

Applying the procedure described in Section 2.6, the grand average across 36 (=18 participants × 2 sleep sessions per participant) in-lab sleep sessions for the estimated DST and the foot distal skin temperature changes with respect to lights-off was calculated and is shown in Figure 7. The hour-long period, intended to capture the falling asleep process, shows a clear increasing trend (as expected) in both curves. The Pearson correlation between these curves is 0.99 ($p<10^{-5}$).

The distal skin temperature values measured on the foot and on the wrist are not identical [5]. This prevents us from directly calculating the agreement between our algorithm estimation of the wrist distal skin temperature and the measured distal foot temperature for the in-lab data. However, comparing the temperature-change dynamics using intent-to-sleep (i.e., lights-off) as a reference allows us to estimate the qualitative agreement between the wrist-based distal skin temperature estimation and foot-based distal skin temperature measurement.

## 4. Discussion

The significant association between skin temperature and sleep, the potentially causal role of the former on the transition from wakefulness to sleep, and sleep maintenance motivates the need for unobtrusive and ecologically valid means to measure the skin temperature while lying in bed. This can enable the characterization of the skin temperature dynamics across a large sample of individuals for a long period of time and under real life conditions.

In this study, we leveraged the capabilities of a smart bed (C360) equipped with temperature sensor strips located on each side of the bed and slightly under the sleeping surface to develop an algorithm to estimate the wrist distal skin temperature. For this purpose, we leveraged the data collected in a study where the participants (owners of C360 smart beds) collected temperature data for fourteen nights sleeping on their own smart bed at home, and for two in-lab nights sleeping on a C360 bed installed in the sleep lab. This design allowed us to train and validate the algorithm with the in-home data and verify the results using the in-lab data.

The relatively slow temporal variation in the skin temperature [15] enabled the resampling of both the sensor strip and the ground-truth data at 1 sample/min. The orientation of the sensor strip (see Figure 1) suggests its sensitivity to lateral sleep movements, which were (in part) compensated by our strategy of selecting, at each time point, the data from the top three out of the five temperature values provided by the sensor.

The architecture of the skin temperature estimation algorithm was organized into two stages; the first stage is a regression model implemented as a gradient boosted decision tree and the second stage leveraged the strong temporal correlation between consecutive skin temperature values to generate the final skin temperature estimation using five consecutive values (i.e., five minutes) from the first stage, which were grouped using a rolling window. We approached the training of this staged regression model in a sequential manner and used a five-fold cross-validation procedure that ensured that the data from a given participant were used either in the training or the validation but not in both. This strategy increases the likelihood that the validation accuracy reflects the generalization accuracy when the model is applied to data from “unseen” individuals.

The convergence of training and validation curves towards a coefficient of determination close to 0.9 (see Figure 4) suggests that the model has an appropriate generalization ability and may explain about 90% of the variability in the reference skin temperature data. The Bland–Altman analysis across all cross-validation folds suggests that the algorithm may not only be able to explain a high proportion of the variance but also reflect the ground-truth skin temperature with a near-zero bias and absolute error <1 °C.

The examination of the data from nine sleep sessions in the validation set (see Figure 6) illustrates the close agreement between the estimated and ground-truth distal skin temperature. It is illustrative to appreciate that the moments where the estimation error appears as more prominent correspond to large deviations in the ground-truth data (see first and last graph in Figure 6). It may be argued that the estimation provided by our algorithm is more stable and as such may be more resilient to perturbations. This is likely the result of the second-stage random forest model that processes five consecutive estimates from the first stage. However, this can also indicate a limitation in the ability to capture rapidly changing temperature variations that may reflect sleep-relevant aspects.

The verification results using the in-lab data provide further evidence of the generalization ability of the algorithm. Indeed, the temperature sensor strip data for each participant were collected from a different smart bed (i.e., in-lab versus at-home), and the distal temperature data were measured on the foot instead of the wrist. Furthermore, there were two more participants for the in-lab condition with respect to the in-home condition. While the distal skin temperature measured on the wrist may not be identical in value to that measured on the foot [5], it can be reasonably expected that during the falling asleep process, both temperatures increase, which facilitates the decrease in the core body temperature. The latter is a hallmark of the transition from wakefulness to sleep [1]. The grand average distal skin temperature change curves shown in Figure 6 show that the foot distal skin temperature increases, which reflects a transition from wakefulness to sleep and that the grand average distal skin temperature estimated by our algorithm shows a similar increasing trend that significantly and strongly correlates (Pearson correlation: 0.99) with the measured distal skin temperature.

To train the distal skin temperature estimation model, strict quality selection criteria were applied. Indeed, about 20% (49/229) of the in-home sleep session data was not considered for training due to its insufficient TSS quality and the data from 85 in-home sleep sessions were also excluded due to the insufficient quality of the ground-truth reference. Emphasizing data quality over quantity for the training of machine learning models has been recommended as a strategy to improve model performance [23]. Notwithstanding the strict data quality criteria applied to model training, the verification with the in-lab data considered all the sleep sessions.

There are a number of limitations associated with this research. The number of participants in this study is small and this limits the generalization of the results we have presented.

The reference for the distal skin temperature was collected from the wrist. However, it may be more appropriate to consider as the distal temperature reference a mean value across several sensors positioned on all limbs. The algorithm architecture proposed in this article may still be able to successfully use that type of reference.

The distal-to-proximal gradient temperature was suggested to be the best predictor for sleep onset [24] and, as such, it may be preferable to design an algorithm to estimate this gradient instead of the distal skin temperature alone.

Sleep stage information from the in-home data is not available given that those recordings did not include electroencephalograms or other PSG signals to enable sleep staging. This limits the analysis of the accuracy depending on the sleep stages.

The precise position of the sleeper on the bed was not recorded, which could have provided valuable context information about the segments with missing or erroneous data that we discarded in our analysis. Our remediation strategy consisted of selecting the top three temperature values as proxies for the sensors that touch the body, and could be optimized if the sleeper location was known.

The accuracy of the system was not tested while using electric blankets or other devices. Provided that these would not directly touch the sensors and modify only the in-bed micro-environment, it is reasonable to expect that the algorithm can still provide accurate skin temperature estimation. However, this needs to be formally validated.

The method proposed in this research to unobtrusively estimate the distal skin temperature while lying in bed has a promising accuracy and generalization ability. To the best of our knowledge, this is the first study reporting on the feasibility of unobtrusively estimating the skin temperature while lying in bed. Additional research is needed to confirm these encouraging results in a larger cohort.

## 5. Patents

G.G.-M. and N.M. are authors in the US Patent Application Bed Having Features for Estimating Core Body Temperatures from Sensing of Cardiac Parameters and External Temperature Patent Application (Application 17/738,818, 2022).G.G.-M. is an author in the US Patent Application Bed Microclimate Control Using Humidity Measurements Patent Application (Application 17/964,150, 2023).G.G.-M., P.C., M.R.R., and N.M. are authors in the US Patent Application Bed System for Adjusting a Sleep Environment Based on Microclimate Temperature and Sleep Quality Optimization (Application 18/370,552, 2024).G.G.-M., P.C., and Y.S. are authors in the US Patent Application for Bed with Bed Presence Detection Using Temperature Signals Patent Application (Application 18/198,431, 2023).

## Figures and Tables

**Figure 1 sensors-24-04882-f001:**
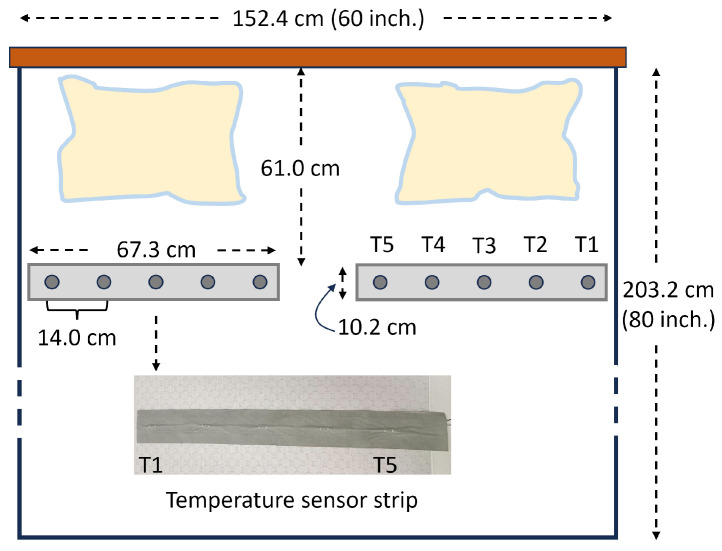
Temperature sensor strip location on the smart bed. This illustration corresponds to the queen-sized bed (width: 152.4 cm, height: 203.2 cm). The inset shows a picture of the temperature sensor strip on a foam layer of the mattress.

**Figure 2 sensors-24-04882-f002:**
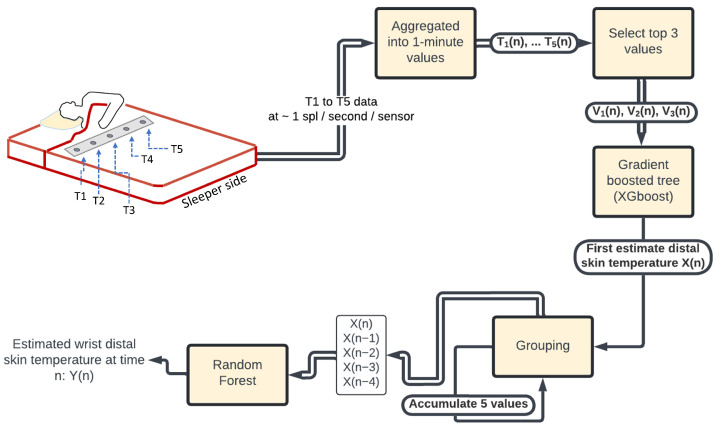
Wrist distal skin temperature estimation algorithm. For convenience of representation, only the sleeper’s side of the bed is illustrated (this is why only the TSS for the sleeper side is visible in this figure). In all diagrams in this article, a block with a yellow background indicates a process and a block with a white background indicates a data unit or model.

**Figure 3 sensors-24-04882-f003:**
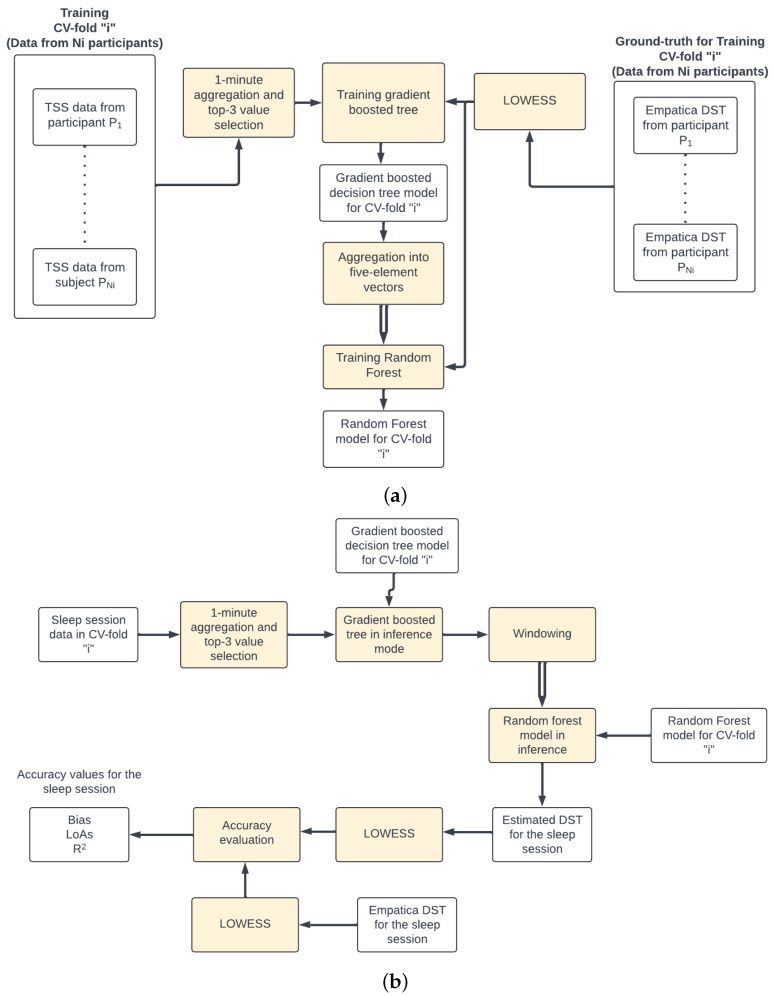
(**a**) Training process for CV fold “i” using the data from Ni participants. (**b**) Accuracy calculation for a sleep session in the validation set of CV fold “i”.

**Figure 4 sensors-24-04882-f004:**
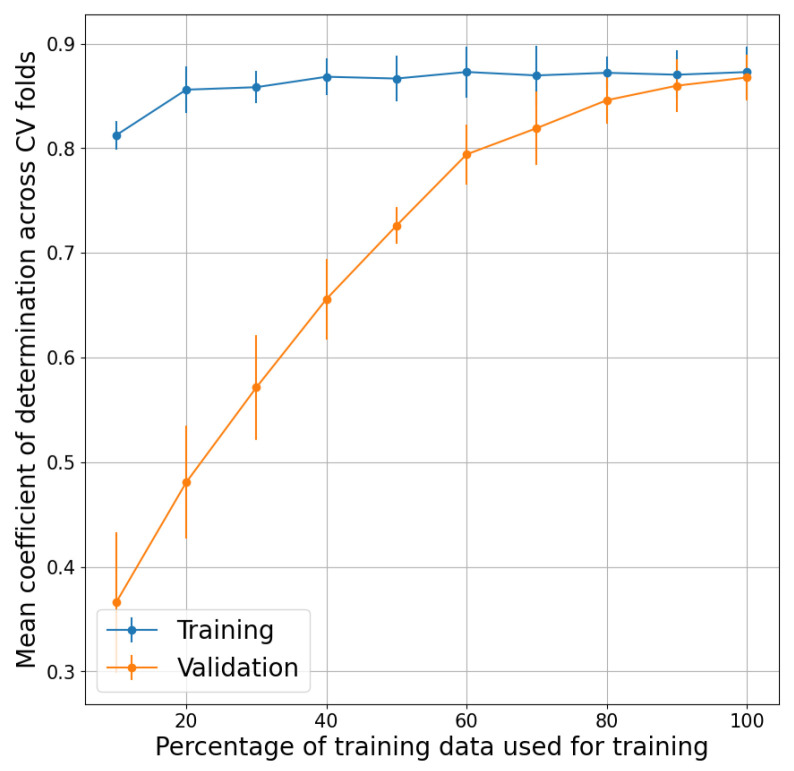
Mean coefficient of determination across the five CV folds versus percentage of training data.

**Figure 5 sensors-24-04882-f005:**
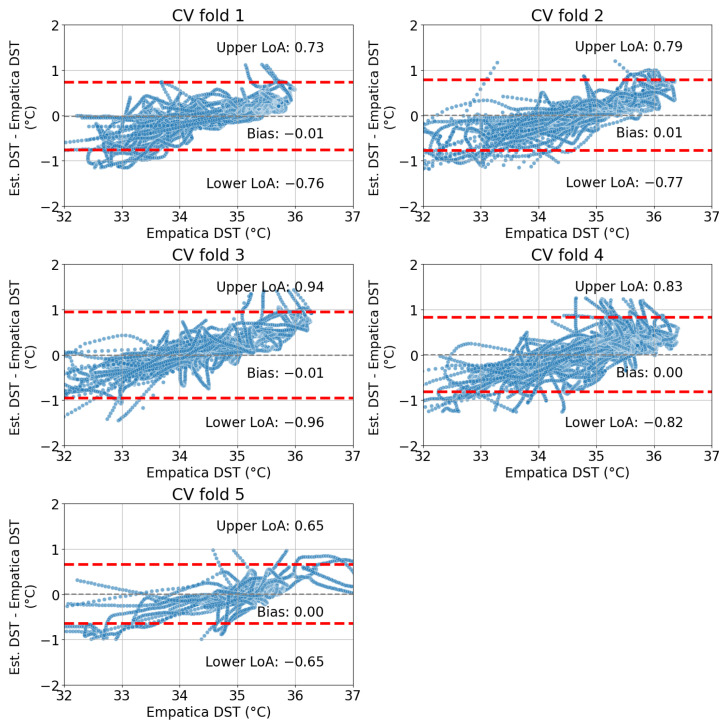
Bland–Altman plots for all CV folds. The red horizontal dashed lines indicate the limits of agreement.

**Figure 6 sensors-24-04882-f006:**
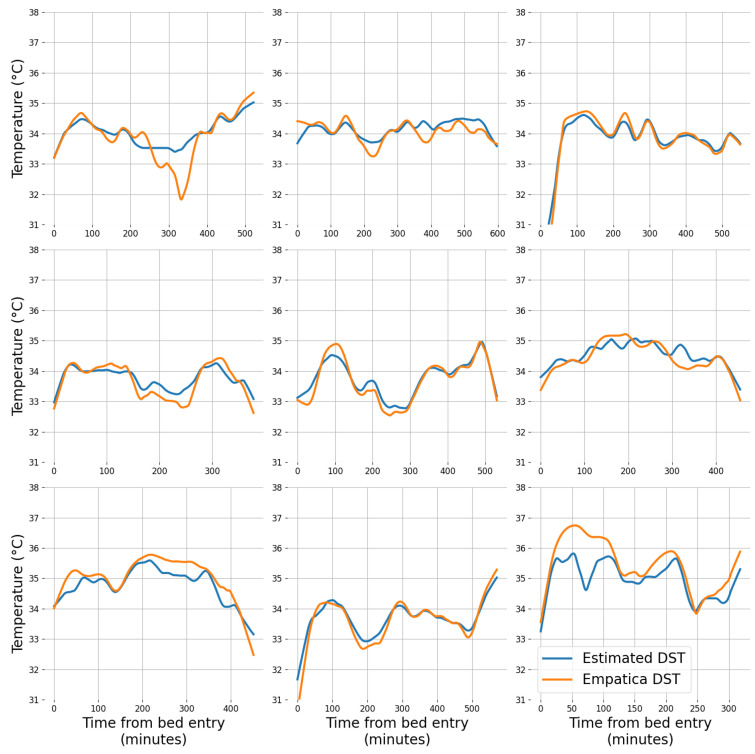
Examples of sleep sessions (from validation set) with ground-truth and estimated DST.

**Figure 7 sensors-24-04882-f007:**
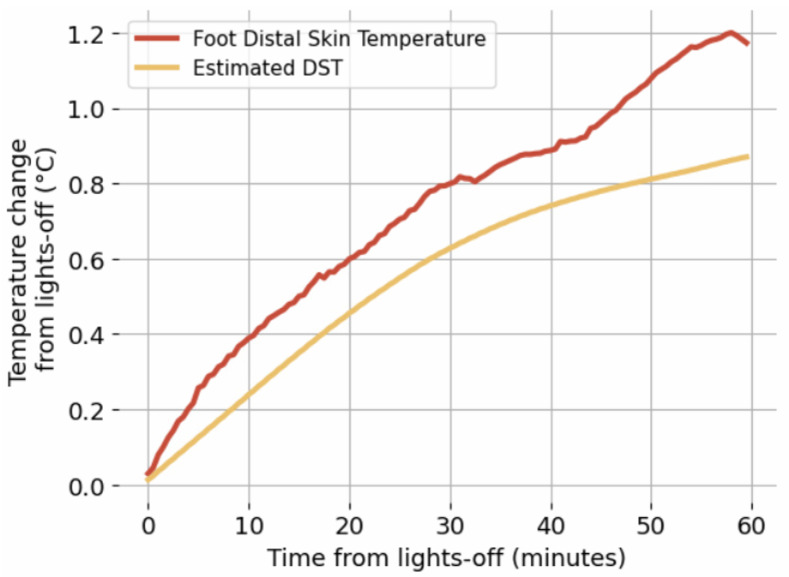
Grand average temperature change curves for estimated DST and foot distal skin temperature. The Pearson correlation between these curves is 0.99.

**Table 1 sensors-24-04882-t001:** Hyperparameter space for the regression models.

Model	Hyperparameter	Value Space
Gradient boosted tree	Number of estimators	100, 200, 300
Gradient boosted tree	Maximum depth	3, 5, 7, 9, 15
Gradient boosted tree	Learning rate	Uniform distribution between 0 and 1.
Random Forest	Bounds for the estimator number	Lower bound 10, upper bound 300.
Random Forest	Bounds for maximum depth	Lower bound 1, upper bound 30.
Random Forest	Bounds for minimum number of samples to split	Lower bound 2, upper bound 20.

**Table 2 sensors-24-04882-t002:** Demographic information of study participants.

ID	Gender	Age	Height (m)	Weight (kg)
S01	Male	41	1.75	112.7
S02	Female	37	1.55	68.7
S03	Male	45	1.83	117.7
S04	Female	39	1.72	68.2
S05	Male	42	1.95	92.9
S06	Female	42	1.61	61.9
S07	Male	43	1.76	87.8
S08	Male	31	1.86	99.6
S09	Female	49	1.57	81.8
S10	Female	49	1.60	79.0
S11	Male	51	1.76	85.5
S12	Female	45	1.60	79.0
S13	Female	57	1.55	81.8
S14	Male	41	1.75	81.8
S15	Male	57	1.79	102.7
S16	Female	42	1.59	84.9
S17	Male	50	1.76	91.7
S18	Female	40	1.73	88.5

**Table 3 sensors-24-04882-t003:** Optimal hyperparameter values for the regression models.

Model	Hyperparameter	Optimum Value
Gradient boosted tree	Number of estimators	300
Gradient boosted tree	Maximum depth	3
Gradient boosted tree	Learning rate	0.031
Random Forest	Number of estimators	300
Random Forest	Maximum depth	unbounded
Random Forest	Minimum number of samples to split	2

**Table 4 sensors-24-04882-t004:** Mean validation accuracy for each CV fold.

CV Fold	Bias (°C)	Lower LoA (°C)	Upper LoA (°C)	Coefficient of Determination $R^{2}$
1	−0.01	−0.76	0.73	0.88
2	0.01	−0.77	0.79	0.89
3	−0.01	−0.96	0.94	0.83
4	0.00	−0.82	0.83	0.87
5	0.00	−0.65	0.65	0.90

## Data Availability

The data analyzed in this article are not available because the approved consent form prevents us from making the data publicly available.

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
