# Peer review of "Unobtrusive Skin Temperature Estimation on a Smart Bed"

_sensors, 2024, doi:10.3390/s24154882_

Round 1

Reviewer 1 Report

Comments and Suggestions for Authors

This paper did an interesting work for estimating the unobtrusive skin temperature based on the signal from the TSS in smart bed. The findings are helpful, and the paper is well written. There are some issues should be treated before the publication.

1.A reference for the smart bed C360 should be added to show its basic information.

2. The relative position between TSS and volunteers may influence the obtained temperature, the height of each volunteer should be added.

3.As shown in 3.1, only 90 sleep sessions (90 from 224) are used for further analysis, more information is needed for the used sessions, such as, maybe, the distribution among volunteers and the sleeping quality, etc. As a research work, it is good to get the required information from limited data. But as a user, it will be depressing when most of obtained data is useless. Therefore, the cause for the appearance of insufficient ones should be discussed, with possible measures to improve the data quality.

4. Fig. 6 gives some temperature data to show the accuracy. It is better to mark the different sleeping stages in it.

5. What about using more TSSs for temperature sensing? Can it improve the results? Will it be compatible with the proposed algorithm?  

Author Response

Comment 1. This paper did an interesting work for estimating the unobtrusive skin temperature based on the signal from the TSS in smart bed. The findings are helpful, and the paper is well written. There are some issues should be treated before the publication.

We thank the reviewer for their positive and encouraging comments about our work. We believe indeed that the temperature sensor strip concept and associated algorithm to unobtrusively estimate skin temperature is relevant and useful for the readership of the Sensors journal.

We are also grateful for the detailed feedback and advice to improve our article. In the following paragraphs we provide our point-by-point response and indicate the corresponding edits we made to the article. To facilitate the identification of the new material we added to the manuscript, we displayed the edits in blue.

We thank in advance the reviewer for their consideration of the revised manuscript.

Comment 2.A reference for the smart bed C360 should be added to show its basic information.

We added a reference to the C360 smart bed at the place where we first mentioned this smart bed model (second paragraph of Section 2.1 “Smart bed platform”).

Comment 3. The relative position between TSS and volunteers may influence the obtained temperature, the height of each volunteer should be added.

The full demographic information for all participants in the study is provided in Table 2 (Section 3.1 Dataset for the analysis).

Comment 4. As shown in 3.1, only 90 sleep sessions (90 from 224) are used for further analysis, more information is needed for the used sessions, such as, maybe, the distribution among volunteers and the sleeping quality, etc. As a research work, it is good to get the required information from limited data. But as a user, it will be depressing when most of obtained data is useless. Therefore, the cause for the appearance of insufficient ones should be discussed, with possible measures to improve the data quality.

This is an important point raised by the reviewer which motivated us to add a new paragraph to the discussion section (page 12; third paragraph; text in blue color). In this paragraph, we acknowledge the rather strict criteria we applied to select the model-training (in-home) data. We added a reference making the case for emphasizing data quality for model training. However, we also mentioned that for the verification (in-lab) phase, we considered all sleep session data without applying data selection criteria.  

Comment 5. Fig. 6 gives some temperature data to show the accuracy. It is better to mark the different sleeping stages in it.

The waveforms in Figure 6, correspond to sleep session data in the validation set which was recorded at home. Unfortunately, we do not have sleep stage information for in-home recordings given that those recordings did not include any EEG or EOG which could have been used for sleep staging. We edited Figure 6’s caption and the text in the article making it clear that the data is from the validation in-home dataset and that sleep stage information is not available for those recordings.

We also added a paragraph in the limitations (highlighted text in green) to make it clear that the absence of sleep stage information is a limitation of this work.

Comment 6. What about using more TSSs for temperature sensing? Can it improve the results? Will it be compatible with the proposed algorithm?

We thank the reviewer for the opportunity to elaborate about the generalization ability of our algorithm. The basic block diagram (in figure 2) can be easily generalized to the case where multiple TSSs are positioned parallel to each other.

Indeed, given the data from N TSSs and assuming that Ti,j refers to the j-th sensor from the i-th TSS, the algorithm would consist in aggregating first the values of each Ti,j to have 1-minute resolution, then select the top three values for each sensor out of the N available ones (to ensure that the considered sensors most likely touch the sleepers’ body).

The next steps in the algorithm, namely the Gradient Boosted Tree, windowing, and random forest can proceed identically to the single TSS case. This leverages the ability of the Gradient Boosted Tree to easily handle larger input sizes and produce a first distal skin temperature estimation which (given the higher number of TSSs) is likely to have higher accuracy compared to the single TSS case. The windowing and random forest processes that follow the Gradient Boosted Tree do not need to change since they will still process a single estimation value.

Reviewer 2 Report

Comments and Suggestions for Authors

Comments to Authors (General)

·       The authors shall have to present high resolution pictures for all figures.

·       The keywords should be at least FIVE.

·       The specific mathematics for the significant portions should be expressed.

Comments to Authors (Specific)

·       The authors shall present the detailed research challenges of present condition in the “Abstract” for the standard of technical writing.

·       The authors shall express the detailed interpretation on Equation (1).

·       The authors shall have to recheck the equation of SDd.

·       The authors shall have to interpret the detail of Figure.5, Figure.6 and Figure.7.

·       The authors shall have to discuss on the impact of interference from all sensors used in this study.

·       The authors shall have to discuss on the windowing function with mathematical expressions in that study.

·       The authors have to present the accuracy and performance of the overall system.

·       The statistic table for performance table of this study by comparing with the recent studies.

·       The recommendations of the developed system must be expressed in conclusion section.

·       Even though some limitations of this study, the effects of physical devices on the testing system shall have to be mentioned in details.

Comments on the Quality of English Language

Moderate editing of English language required

Author Response

Comments to Authors (General)

Comment 1. The authors shall have to present high resolution pictures for all figures.

We thank the reviewer for this recommendation. As part of the revised submission, we uploaded high-resolution figures as separate files.

Comment 2. The keywords should be at least FIVE.

We agree with the reviewer and have added two keywords (for a total of six keywords) that are relevant for this manuscript’s theme: sleep, and regression model

Comment 3. The specific mathematics for the significant portions should be expressed.

We thank the reviewer for the suggestion. We added mathematical expressions in the 2nd paragraph in page 4 to clarify the selection of the top 3 values, in the 3rd paragraph in page 4 to clarify the estimation of X(n), and also to clarify the meaning of Y(n). We have also expanded on the explanation of Equation 1 and added mathematical expressions when applicable.

Comments to Authors (Specific)

Comment 4. The authors shall present the detailed research challenges of present condition in the “Abstract” for the standard of technical writing.

We agree with the reviewer that we need to make it clear in the Abstract the research challenges that remain to be tackled. The small sample size is the most important one to address in our future research efforts. We edit the Abstract section accordingly.

Comment 5. The authors shall express the detailed interpretation on Equation (1). The authors shall have to recheck the equation of SDd.

We agree with the reviewer comment on providing additional information to interpret Equation1 and make the article self-contained. We expanded the description of Equation 1 and corrected a typo in the definition of SDd.

Comment 6. The authors shall have to interpret the detail of Figure.5, Figure.6 and Figure.7.

We followed the recommendation from the reviewer and expanded the description for Figure 5 to 7. The added text in the manuscript is highlighted in blue. Additional qualitative interpretation for these figures can be found in the discussion section.

Comment 7. The authors shall have to discuss on the impact of interference from all sensors used in this study.

This is an interesting point to elaborate on.

First from an electromagnetic (EM) perspective, the degree of interference is minimal because of the intrinsic slow dynamics of the skin temperature signal that require a low sampling rate which makes the signal of interest less vulnerable to EM interference that typically occurs at higher frequencies. In addition, the temporal aggregation and smoothing that we applied to have 1-minute resolution helps increase the signal-to-noise ratio.

Second, from an inter-sensor interference perspective it is important to mention that the individual sensors in the TSS operate independently. While it is not possible to guarantee that all the sensors have exactly the same sampling frequency (1 Hz), we do not require such a high sampling frequency because we sub-sample at 1-minute periodicity.

Finally, the fact that we select the top three values at each minute helps us in maximizing the information provided by each individual sensor and minimize the potentially non-informative components from sensors that do not touch the body at a specific time.

Comment 8. The authors shall have to discuss on the windowing function with mathematical expressions in that study.

We thank the reviewer for noticing this. We are not applying any windowing function but a simple grouping of 5 values X(n-4) to X(n). To remove this confusion, we edited the diagram in Figure 2 and replaced the word “Windowing” by “Grouping” to make it consistent with the text.

Comment 9. The authors have to present the accuracy and performance of the overall system.

We agree with the reviewer. Table 4, Figures 5 to 7 represent the accuracy of the overall system.

Comment 10. The statistic table for performance table of this study by comparing with the recent studies.

Thank you for this suggestion. To the best of our knowledge this is the first research reporting on the feasibility of unobtrusively estimating skin temperature while lying in bed. We have added this to the discussion section.

Comment 11. The recommendations of the developed system must be expressed in conclusion section.

We agree with the reviewer and we added this as last paragraph in the article.

Comment 12. Even though some limitations of this study, the effects of physical devices on the testing system shall have to be mentioned in details.

We agree with the reviewer and added a paragraph about this in the discussion section.

Reviewer 3 Report

Comments and Suggestions for Authors

The manuscript entitled “Unobtrusive skin temperature estimation on a smart bed” is interesting, and the authors proposed a two-stage regression model to estimate distal skin temperature based on a temperature sensor strip. However, the paragraph and expression of this manuscript need well-organized, and the quality of the figures needs improving.

 (1) Page 59 on Page 2, what are the basic parameters or measurement mechanisms of the temperature sensors in the C360 bed, and iButton and Empatica E4?  The basic sensitivity and resolution of the sensors should be provided in Materials and Methods.

(2) On page 3, is there any difference between One strip on a bed and two strips on a bed? The models about strip location shown in Figure 1 and Figure 2 are different, actually, the volunteers independently participated in one bed for one person, and two strips on a bed confused the experiments and increased the variables.

(3) The quality of Figure 4 and Figure 5 should be improved, for example, there is no X and Y axis in Figure 4 and Figure 5, the scale of the Y axis in Figure 5 is a little bit large and some labels in the X axis are missing.

(4) Line 113 on Page 4, why the top three largest values were selected from the five TSS values, rather than the mean? Although the body most likely touched these sensors, please explain the reasonability and efficiency of this method.

(5) On page 11, the authors provided the temperature change curves from lights-off in Figure 7 based on the in-lab foot distal skin temperatures, the accuracy curve of the validation should be presented for comparison and further analysis.

Comments on the Quality of English Language

to meet the criterion of scientific writing, the paragraph and expression of this manuscript need well-organized, and the quality of the figures needs to improve.

Author Response

Comment 1. The manuscript entitled “Unobtrusive skin temperature estimation on a smart bed” is interesting, and the authors proposed a two-stage regression model to estimate distal skin temperature based on a temperature sensor strip. However, the paragraph and expression of this manuscript need well-organized, and the quality of the figures needs improving.

We thank the reviewer for the positive appreciation of our manuscript. We thoroughly revised the manuscript according to recommendations from reviewers, highlighted the edits we made in blue color, submitted high resolution figures as separate files, and resubmitted the revised manuscript. We thank in advance the reviewer for their consideration of the revised manuscript.

Comment 2. Page 59 on Page 2, what are the basic parameters or measurement mechanisms of the temperature sensors in the C360 bed, and iButton and Empatica E4?  The basic sensitivity and resolution of the sensors should be provided in Materials and Methods.

We agree with the reviewer recommendation. We added these details in Section 2.1 for the TSS and in Section 2.2 for iButton and Empatica.

Comment 3.  On page 3, is there any difference between One strip on a bed and two strips on a bed? The models about strip location shown in Figure 1 and Figure 2 are different, actually, the volunteers independently participated in one bed for one person, and two strips on a bed confused the experiments and increased the variables.

We thank the reviewer for noticing this. All the beds have two TSSs; on each side of the bed. Figure 2 illustrates only the sleeper side of the bed for convenience of representation. We modified Figure 2 and its caption to make it clear that only the sleeper side of the bed was shown.

Comment 4.  The quality of Figure 4 and Figure 5 should be improved, for example, there is no X and Y axis in Figure 4 and Figure 5, the scale of the Y axis in Figure 5 is a little bit large and some labels in the X axis are missing.

We agree with the reviewer about the need for these changes. We modified the figures as requested by the reviewer.

Comment 5.  Line 113 on Page 4, why the top three largest values were selected from the five TSS values, rather than the mean? Although the body most likely touched these sensors, please explain the reasonability and efficiency of this method.

This is an interesting and relevant point. Taking the mean across all five sensors for each time point results in:

  • Loss of spatial information because we would be collapsing temperature-gradient-type of information into a single value (e.g. values across the 5 temperature sensors of for instance 28 °C, 32 °C, 33 °C, 30 °C, 32 °C would get reduced to a single value of 31 °C). Such value is not informative enough for skin temperature estimation as seen in the figure below where the result (in green) of using the mean value across T1 to T5 was used to estimate distal skin temperature instead of the algorithm proposed in our manuscript (in blue). The reference waveform is represented in orange. These results suggest that taking the top three values is preferable to taking the mean across T1 to T5.
  • The addition of uninformative data from sensors that are not touching the body. This can decrease the performance of the system (Rothacher, Y., & Strobl, C., 2023).

Rothacher, Y., & Strobl, C. (2023). Identifying Informative Predictor Variables With Random Forests. Journal of Educational and Behavioral Statistics, 0(0). https://doi.org/10.3102/10769986231193327

Comment 6. On page 11, the authors provided the temperature change curves from lights-off in Figure 7 based on the in-lab foot distal skin temperatures, the accuracy curve of the validation should be presented for comparison and further analysis.

The distal skin temperature values measured on the foot and on the wrist are not identical. This prevents us from directly calculating the agreement between our algorithm estimation of wrist distal skin temperature and the measured distal foot temperature for in-lab data. However, comparing the temperature-change dynamics using intent-to-sleep (i.e., lights-off) as reference allows us to estimate the qualitative agreement (characterized by a rather high Pearson correlation of 0.99) between wrist-based distal skin temperature estimation and foot-based distal skin temperature measurement. Following the reviewer’s recommendation, we added a paragraph in section 3.3 (second paragraph) elaborating on this matter. The second paragraph in page 12, discusses this aspect as well.

Round 2

Reviewer 1 Report

Comments and Suggestions for Authors

It is a good revision, and all my comments have been addressed.

In my opinion, this paper can be accepted now.